# Polymer-constrained excimer enables flexible and self-healable optoelectronic elastomer for mechanical sensor

Shuyu Zheng[1,2], Dazhe Zhao [3], Nengjie Cao[1], Jiajia Zhou [1,2] ✉,
Junwen Zhong [3] ✉ & Haobing Wang [1,2] ✉

The development of high-performance, flexible, and self-healable optoelectronic materials is pivotal for advancing next-generation wearable technologies. In this study, we introduce nanoscale naphthyl-naphthyl microphase separation into a polyisoprene matrix, endowing olefin copolymers with exceptional mechanical properties, high flexibility, and intrinsic self-healing capabilities at room temperature without external stimuli. Notably, by employing a "polymer-constrained excimer" strategy, these copolymers exhibit remarkable photoluminescent properties, achieving an ultra-high photoluminescence quantum yield (PLQY > 98%) through the formation of naphthyl-naphthyl excimers. Experimental and theoretical analyses reveal that under the encapsulation of flexible *cis*-1,4-polyisoprene segments, nanoscale naphthyl aggregates form stable excimers upon UV stimulation, resulting in extraordinary fluorescence quantum efficiency. Additionally, the nanoscale aggregation of naphthyls imparts superior electret performance to these copolymers, making them ideal for opto-electro-mechanical sensors for the robotic hand and other devices.

The demand for high-performance, flexible, and self-healing optoelectronic materials has surged with the rise of next-generation wearable technologies, such as electronic skin, smart sensors, and humanoid robots[1–11]. These applications require materials that are lightweight, durable, and capable of capturing, analyzing, and conveying multidimensional signals (e.g., optical and electrical) while providing high-precision feedback[12–14]. However, traditional high-performance materials often suffer from high rigidity or require good dispersion in non-functional matrices, compromising their efficiency and durability in wearable devices. Thus, developing materials that combine mechanical flexibility, intrinsic self-healing properties, and high optical/electronic sensitivity remains a significant challenge[15,16].

Polyolefin thermoplastic elastomers (TPEs), known for their low density, cost-effectiveness, flexibility, and potential for self-healing, offer a promising platform for wearable integration[17–25]. For instance, Hou et al. recently demonstrated that homopolymerization of isoprene by rare-earth catalyst could yield sequence-controlled polyisoprenes with microphase-separated nanodomains of short 3,4-polyisoprene blocks (hard segments) within a flexible *cis*-1,4-polyisoprene matrix (soft segments), exhibiting excellent elasticity and self-healability[26]. However, the limited optical and electronic functionality of non-polar polyolefins restricts their application in optoelectronic devices. To address this, incorporating functional moieties into polyolefin matrices has been explored[27–31], but achieving high

[1]Advanced Institute for Soft Matter Science and Technology (AISMST), School of Emergent Soft Matter, State Key Laboratory of Pulp and Paper Engineering, South China University of Technology, Guangzhou, China. [2]Guangdong Provincial Key Laboratory of Functional and Intelligent Hybrid Materials and Devices, Guangdong Basic Research Center of Excellence for Energy and Information Polymer Materials, South China University of Technology, Guangzhou, China. [3]Department of Electromechanical Engineering and Centre for Artificial Intelligence and Robotics, University of Macau, Macau SAR, China. ✉e-mail: zhouj2@scut.edu.cn; junwenzhong@um.edu.mo; haobingwang@scut.edu.cn

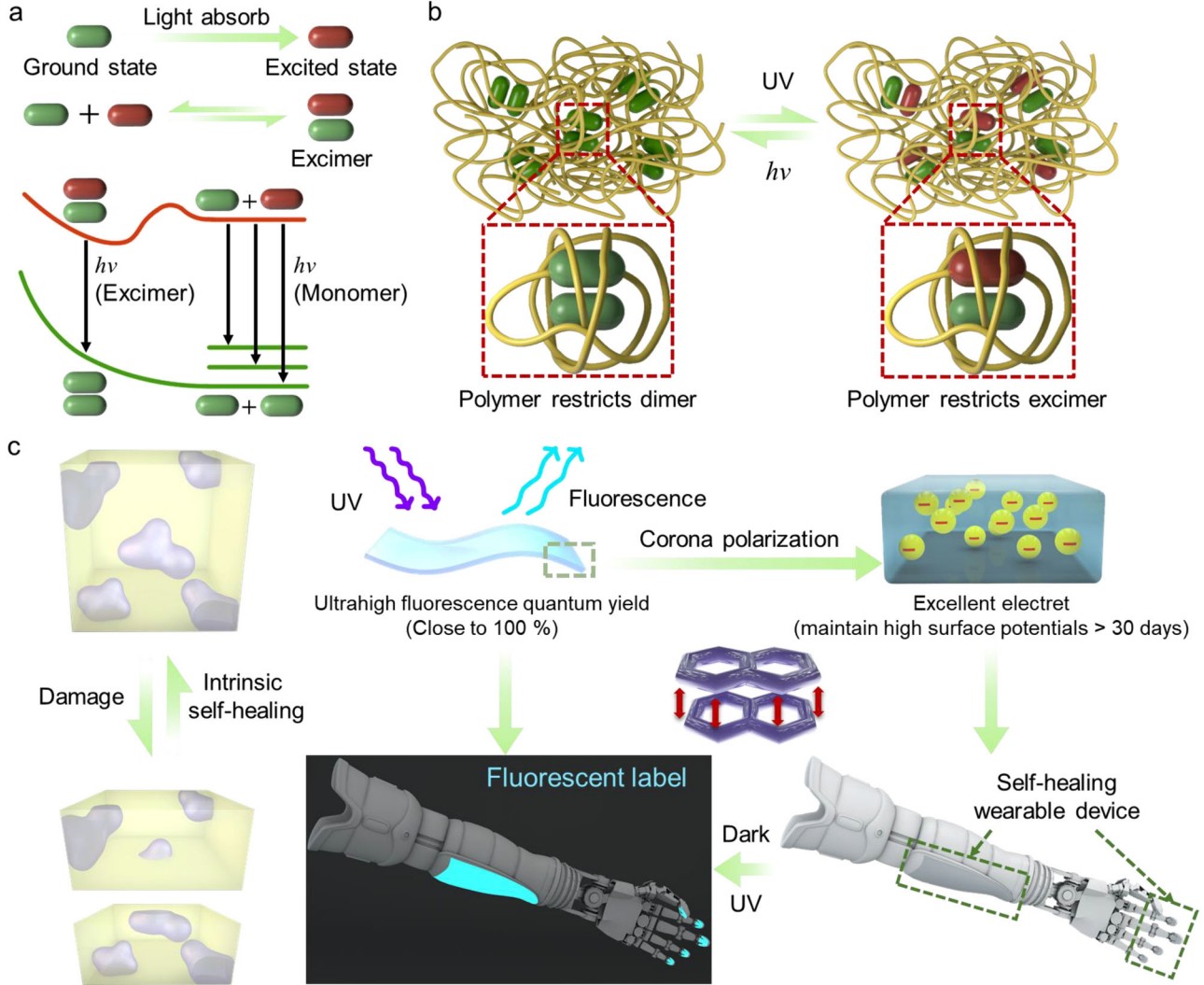

**Fig. 1 | Molecular design and multifunctional integration of isoprene-*co*-1-vinylnaphthalene copolymers. a** *Mechanism of excimer formation*: The π-π stacking interaction between adjacent naphthalene groups enables the formation of excited-state dimers (excimers), leading to red-shifted fluorescence emission compared to monomeric chromophores. **b** *The "polymer-restricted excimer" strategy*: By precisely controlling the spacing and mobility of naphthalene side chains through copolymer architecture, the excimer formation is confined within the polymer matrix. **c** *Integration of multifunctional properties*: The copolymer system simultaneously achieves (i) *self-healing* via reversible physical crosslinking, (ii) *high fluorescence efficiency* due to optimized excimer emission, and (iii) *electret behavior* through charge trapping at the naphthalene-rich microdomains.

photoluminescence efficiency alongside self-healing properties remains elusive.

Introducing aromatic chromophores into polymer backbones can enhance optoelectronic functionality[32–37], which is valuable for optical sensing, biomedical imaging, and energy conversion[38–42]. Precise control of intermolecular spacing of aromatic units is critical, as excessive overlap can lead to aggregation-caused quenching (ACQ)[43–46]. Conversely, appropriate aromatic-aromatic interactions may also facilitate excimer formation[47–51], thereby enhancing fluorescence quantum yields of materials (Fig. 1a). However, traditional excimer formation typically requires rigid scaffolds or solvent dispersion[52–56], limiting its application in flexible wearable materials.

Aromatic-containing polymers also constitute an important category of polymer electrets. These materials exhibit a distinctive characteristic that sets them apart from most alternatives: the innate capacity to produce stable electrostatic charges through mechanical stimulation (e.g., vibration, compression) in the absence of external polarization fields[57]. This unique attribute makes them particularly valuable for applications ranging from acoustic transducers to mechanical energy harvesting systems. Beyond their electret

properties, these polymers offer additional advantages including intrinsic flexibility, robust mechanical durability, and excellent compatibility with organic and soft polymeric substrates-features that clearly differentiate them from conventional inorganic electret materials. Such exceptional properties have generated growing research interest in their application for next-generation flexible electronic devices, including conformal electromechanical sensors, epidermal electronic systems, and tactile feedback interfaces[58–62]. Nevertheless, significant challenges persist in developing intrinsically compliant electret materials. Notably, the integration of advanced functional characteristics - especially self-healing mechanisms to improve service lifetime - represents an entirely unexplored research frontier in soft electret technology.

In this study, we synthesize a family of 1-vinylnaphthalene and isoprene copolymers with varying microstructures and sequences. By employing the formation of nanoscale naphthyl-naphthyl microphase separation in a polyisoprene matrix, these copolymers exhibit a rare combination of mechanical robustness and intrinsic self-healing capabilities at room temperature without external stimuli. The polyisoprene segments constrain naphthyl-naphthyl nanophases, enabling

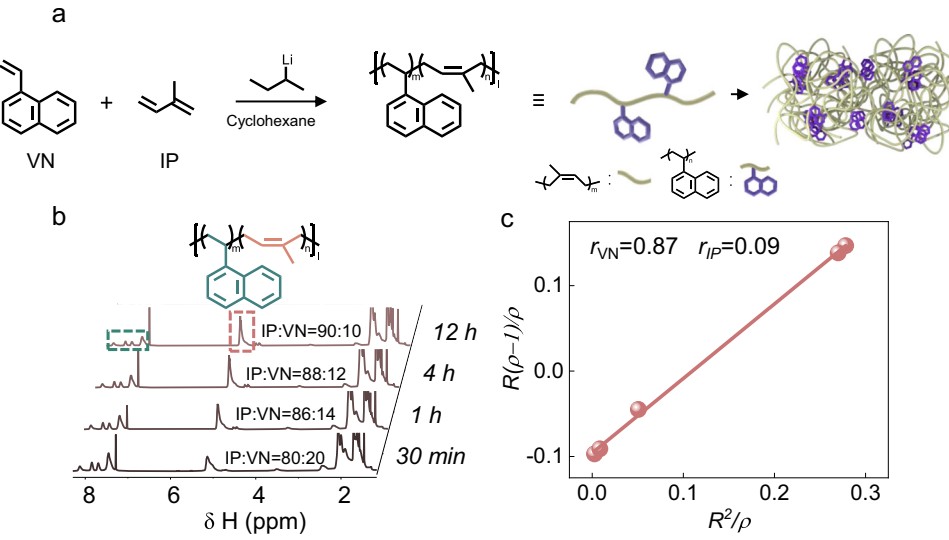

**Fig. 2 | Synthesis of VN-*co*-IP (PNI). a** Copolymerization of IP and VN. **b** [1]H NMR spectra tracking copolymerization of IP and VN (CDCl₃, 400 MHz). **c** Reactivity ratio plot for IP and VN during copolymerization.

the formation of stable excimers (Fig. 1b). Under UV stimulation, this "polymer-constrained excimer" strategy enables the copolymer with an ultra-high photoluminescence quantum yield (PLQY > 98%) in the solid state. Furthermore, the naphthyl-naphthyl nanophase demonstrates exceptional electret performance with over 30-day charge retention, expanding its potential applications in self-healable opto-electro-mechanical sensing for the robotic hand (Fig. 1c).

## Results

### Synthesis of copolymers

The copolymerization of isoprene (IP) with vinyl monomers such as styrene (St) and 1-vinylnaphthalene (VN) was initiated by *sec*-butyl-lithium (*s*-BuLi) in cyclohexane (Fig. 2a, Supplementary Figs. 1, 2). For IP: St (feed ratio 90: 10), tapered block copolymers formed, with IP**:** St ratios of 98: 2, 98: 2, 96: 4, and 90: 10 at 30 min, 1 h, 4 h, and 12 h, respectively[63]. In contrast, real-time monitoring of IP/VN copolymerization (feed ratio 90: 10) revealed simultaneous incorporation of both monomers, with IP: VN ratios of 80: 20, 86: 14, 88: 12, and 90: 10 at the same time, suggesting a random copolymerization (Fig. 2b). The reactivity ratios of this copolymerization were calculated as $r_{IP} = 0.09$ and $r_{VN} = 0.87$, which further confirmed that the copolymerization took place in a random fashion (Fig. 2c). As the [IP]/[VN] feed ratio was decreased from 90/10 to 80/20, 75/25, 70/30, 65/35, 60/40, 50/50, and 30/70, the [IP]/[VN] ratio of the resulting copolymers showed a similar trend from 90/10 to 30/70, respectively (Supplementary Table 1 and Figs. 3, 4). The ¹³C NMR and DOSY NMR spectra in CDCl₃ demonstrated the formation of the random copolymer of poly-(1-vinylnaphthalene)-*random*-(*cis*-1,4-isoprene) (PNI) rather than the mixture of two homopolymers (Supplementary Figs. 5, 6).

### Mechanical and self-healing properties of PNI copolymer

The VN-*co*-IP copolymer (PNI) with 10 mol% VN content exhibited a glass-transition temperature ($T_g$) at −45 °C (Fig. 3a and Supplementary Fig. 7). As the VN content increased from 10 mol% to 70 mol%, the copolymers showed incremental glass-transition temperatures ($T_g$) from −45 °C to −30 °C, 10 °C, 40 °C, 70 °C, and 100 °C (Fig. 3a), respectively. These copolymers with different VN contents showed distinct mechanical properties at room temperature. For instance, when the content of VN was lower than 20 mol%, the copolymer exhibited very weak tensile strength (<0.1 MPa) and poor elasticity. When the content of VN was higher than 40 mol%, the copolymers showed much higher $T_g$ (Fig. 3a) and behaved like hard plastics with

high Young's modulus (100 MPa) and tensile strength at break (25 MPa, Fig. 3b). The optimal copolymer with approximately 30 mol% VN content ($T_g = 5$ °C) showed typical features of the elastomer with Young's modulus of 0.5 ± 0.1 MPa, breaking strength of 1 ± 0.1 MPa, and elongation at break of 2020 ± 20% (Supplementary Table 2).

With the similar IP: VN ratios of approximately 70: 30, the higher molecular-weight copolymers ($M_n = 117, 143, 184, 201$, and 227 kDa, code as P1-P5, Table 1 and Supplementary Fig. 4) demonstrated the higher tensile strength (2.4 ± 0.1, 3 ± 0.5, 12 ± 0.5, 20 ± 1 and 25 ± 1 MPa) and excellent stretchability with elongations of 1200% to 1950% (Fig. 3c and Supplementary Table 2). In addition to excellent elasticity (Supplementary Fig. 8), these pristine copolymers showed remarkable self-healability. For instance, when a dog-bone-shaped film sample of P3 was completely cut by a razor blade, the material could be stretched to 550% after the cut area was contacted and gently pressed for less than 15 s at 25 °C in air, and then allowed to self-repair for 30 min (Fig. 3d). After healing for 48 h, the fracture healed almost completely as evidenced by observation of a comparable elongation with that of the pristine sample, its original tensile strength of 11 MPa with an elongation of 1270%. When the sample was cut, contacted, and placed under 36 °C (human body temperature) in air, a complete recovery was observed within 6 h, indicating its high potential for self-healing wearable devices or electronic skin (Supplementary Fig. 9c). The picture of a square hot-pressing P3 sheet (dumbbell-configured specimens according to JIS K-6251-7; width: 2 mm; length: 12 mm; thickness: 1 mm) and a movie of a tiny pocket made by P3 full of water are shown in Fig. 3e and Supplementary Movie 1 to present the self-healing effect visually. Tougher copolymers with higher molecular weight, such as P5, could also reach their original tensile strength of as high as 24.5 MPa with an elongation of 1200%, albeit with a longer healing time (7 days) at room temperature (Supplementary Fig. 9a).

When a film sample of P3 was cut by a razor blade, the crack became almost invisible within 1 min at room temperature in the air without any external intervention (Fig. 3f; see also Supplementary Movie 2), standing in sharp contrast with the VN ratios of 50 mol%, which presented no self-healability under the same conditions. As for PNI with 10 mol%, the sample showed limited self-healability and was not able to completely heal the wound in 30 min. A triblock copolymer of VN-*b*-IP-*b*-VN with a total VN content of 30 mol% (NIN-3) also showed no self-healability under the same conditions (Fig. 3f), demonstrating that the self-healing property of the copolymers significantly depends on their microstructures (Supplementary Figs. 10–14). We propose

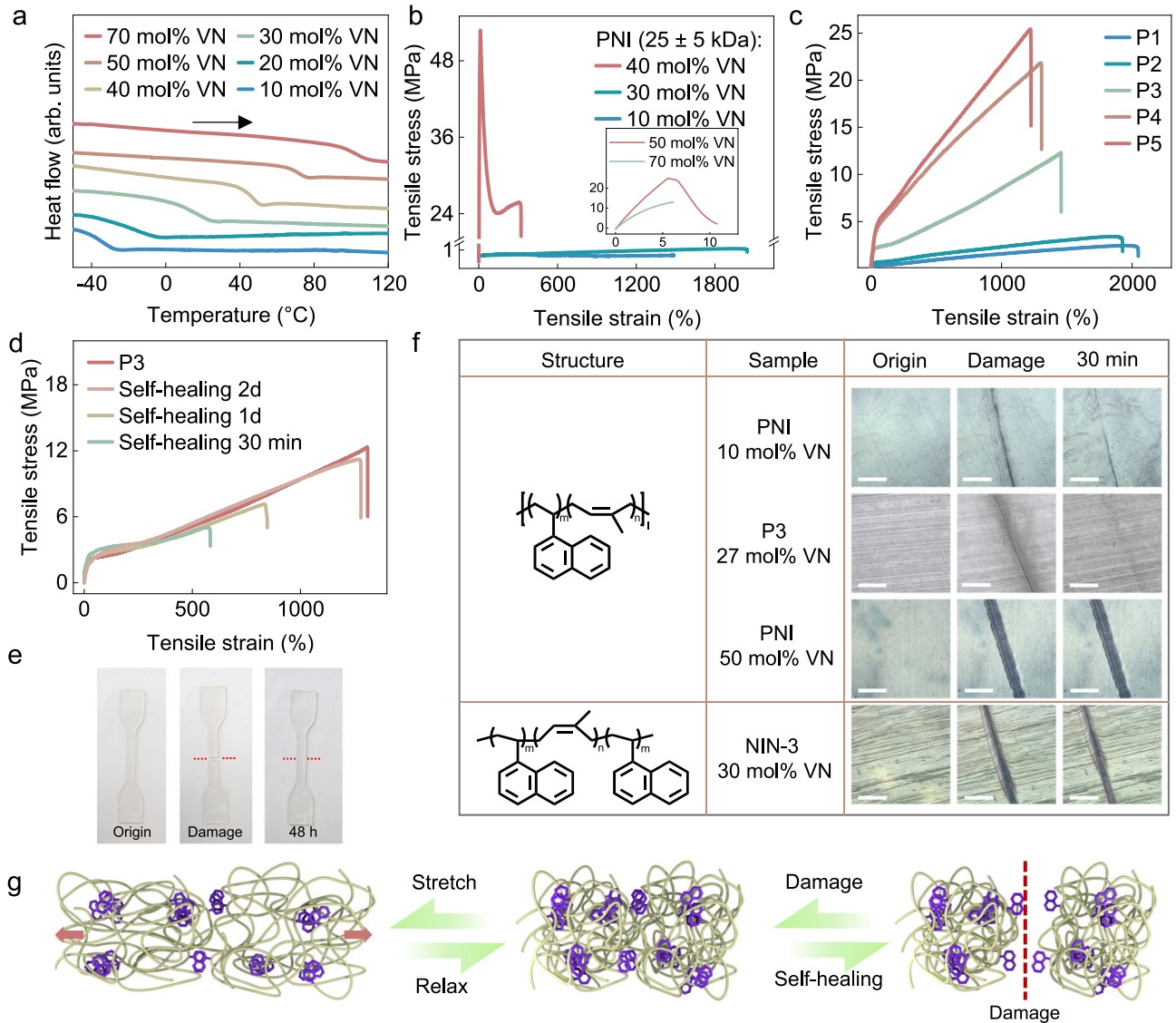

**Fig. 3 | Mechanical and self-healing properties of VN-*co*-IP (PNI). a** DSC analysis (use the data from the second heating, and the endothermic direction is from top to bottom) of PNI with varying VN content (10–70 mol%). **b** Stress-strain curves of PNI copolymer with different VN contents at low $M_n$ (25 ± 5 kDa). **c** Stress-strain curves of P1-P5 ($M_n$ = 117 to 227 kDa, IP/VN ratios = ~70/30). **d** Self-healing tests of P3. **e** P3 film with different states: original, completely severed, and fully self-healed after 48 h (dumbbell-configured specimens according to JIS K-6251-7; width: 2 mm; length: 12 mm; thickness: 1 mm). **f** Optical microscopy of healed PNI films (scale bar: 200 μm). **g** Proposed mechanism for the mechanical and self-healing properties of PNI copolymer.

that the microphase-separated naphthyl nanodomains act as cross-linking points within the flexible *cis*-1,4-polyisoprene matrix, thus enforcing elasticity and toughness (Fig. 3g). The exceptionally rapid self-healing may result from the high mobility of the *cis*-1,4-polyisoprene segments with low $T_g$, which could induce the rapid re-aggregation of the naphthyl groups through π-π interaction to repair the mechanical damage.

## Optical property of PNI copolymer

P3 prepared by hot pressing demonstrated excellent optical transparency in the visible spectrum (400–800 nm), achieving transmittance values of 82-92% (Fig. 4a) due to its amorphous nature (Fig. 3a). Interestingly, under ultraviolet light irradiation, the colorless P3 sample exhibited excellent photoluminescent performance (Fig. 4b), demonstrating a fluorescence quantum efficiency >98% (Fig. 4c and Supplementary Fig. 15). In contrast, both naphthalene (Naph) dispersed in polyisoprene (30 mol%) and THF solution (1 M) didn't show

any notable fluorescence performance (Fig. 4b). We investigated the fluorescence behavior during the self-healing process of P3 (Supplementary Fig. 16). Immediately after cutting and rejoining the P3 specimen, we observed a significant fluorescence enhancement. This phenomenon likely resulted from increased light absorption at the freshly exposed wound surfaces compared to intact regions, where excitation light was attenuated by the penetration process and thus generated less fluorescence. As self-healing progressed, the fluorescence intensity gradually decreased, eventually returning to near-baseline levels upon complete restoration of the material.

To investigate the fluorescence mechanism, we examined the fluorescence property of P3. With a concentration of 0.2 g/L P3 in THF, it exhibited a broad emission range of 310–500 nm with maximum peaks at 340 and 400 nm (Fig. 4d). In contrast, the solid P3 film exhibited a much stronger emission in the range of 340–500 nm with maximum peaks at 396 nm (Fig. 4e), possibly caused by the formation of naphthyl-naphthyl excimers in P3. To verify this hypothesis, the

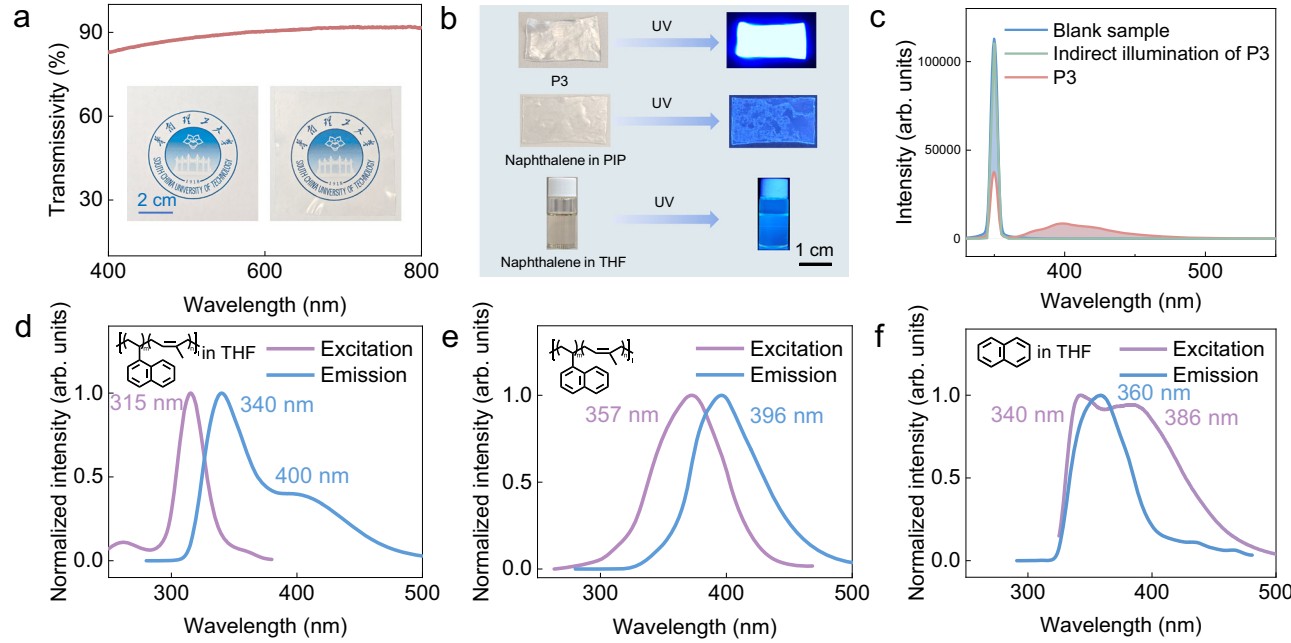

**Fig. 4 | The optical properties of VN-*co*-IP (PNI). a** Visible light transmittance spectrum of P3. **b** Photoluminescence under UV light: P3 (solid), Naph/PIP, Naph in THF (1 M). **c** PLQY measurement of P3. **d**–**f** Fluorescence spectra of: **d** P3 (THF, 0.2 g/L), **e** P3 (solid), **f** Naph in THF (1 M).

**Table 1 | Copolymerization of isoprene (IP) and 1-vinylnaphthalene (VN)[a]**

| Code | [IP]: [VN]: [s-BuLi][b] | conv. IP (%) | conv. VN (%) | $M_n$ (×10³ g mol⁻¹)[c] | IP/VN[d] | $M_w/M_n$[c] | $T_g$ (°C)[e] |
|---|---|---|---|---|---|---|---|
| P1 | 1500: 600: 1 | 96% | 82% | 117 | 73/27 | 1.1 | 10 |
| P2 | 1800: 720: 1 | 95% | 82% | 143 | 72/28 | 1.6 | 10 |
| P3 | 2000: 800: 1 | 92% | 79% | 184 | 73/27 | 2.5 | 11 |
| P4 | 2200: 880: 1 | 90% | 78% | 201 | 73/27 | 2.3 | 13 |
| P5 | 2500: 1000: 1 | 90% | 77% | 227 | 73/27 | 2.3 | 15 |
| P6 | 2500: 300: 1 | 90 % | 83 % | 201 | 90/10 | 1.5 | −40 |
| **P7** | 800: 950: 1 | 77 % | 65 % | 86 | 50/50 | 3.6 | 70 |

[a]Conditions: Sec-Butyllithium (s-BuLi) (1.3×10⁻² mmol); 50 mL Cyclohexane; Room temperature.
[b]Feed ratio (in moles) of isoprene (IP), 1-vinylnaphthalene (VN), and s-BuLi.
[c]Determined by gel permeation chromatography (GPC) in tetrahydrofuran (THF) at 35 °C against polystyrene standard. $M_n$ = number-average molecular weight, $M_w$ = weight-average molecular weight.
[d]Molar ratio of isoprene (IP) and 1-vinylnaphthalene (VN) in the copolymer, determined by ¹H nuclear magnetic resonance (NMR) analysis.
[e]Determined by differential scanning calorimetry (DSC).

fluorescence spectra of naphthalene in THF with different concentrations were recorded (Fig. 4f and Supplementary Fig. 17). The diluted Naph solution (1.0 mM) exhibited excitation in the range of 280–380 nm with maximum peaks at 307 nm and 359 nm and an emission in the range of 310–360 nm with a maximum peak at 330 nm (Supplementary Fig. 17a). Upon increasing the Naph concentration to 1.0 M, both excitation and emission showed a significant red shift in the range of 325–480 nm with maximum peaks at 340 nm and 386 nm and in the range of 320–480 nm with maximum peak at 360 nm, respectively (Fig. 4f), consistent with the optical behavior of Naph excimers in the solution state[47–49]. The fluorescence lifetime is also key evidence for the formation of excimers. When the concentration of Naph in THF increased from 1.0 mM to 1.0 M, the fluorescence lifetime increased from 0.47 ns to 1.11 ns. Remarkably, in comparison to the copolymer in solution (fluorescence lifetime: 1.11 ns), the fluorescence lifetime of P3 in the solid state significantly increased to 5.09 ns (Fig. 5a). The temperature-dependent fluorescence spectroscopy of P3 also showed a significant fluorescence intensity increase as the temperature decreased (Fig. 5b) due to the further reduction of non-

radiative decay; these two phenomena further confirmed the existence of stable excimers in the solid state.

The VN content significantly impacted the fluorescence quantum yield of the copolymer. As the VN content increased from 10 mol% to 50 mol%, the fluorescence quantum efficiency of the copolymers initially increased from 45% (10 mol% VN), reached a peak at 98% (30 mol% VN) and then decreased to 18% (50 mol% VN) (Fig. 5c and Supplementary Fig. 18). The decrease in fluorescence quantum yield at low VN content (e.g., 10 mol%) may be attributed to a lower probability of excimer formation. In contrast, when the VN content was higher (e.g., 50 mol%), the aggregation of numerous naphthyl groups within the copolymer caused a phenomenon of aggregation-induced fluorescence quenching, which reduced the quantum yield of the copolymer. Additionally, the quantum yields of Naph with different molar ratios (10 mol% to 50 mol%) dispersed into the polyisoprene (PIP) were recorded, and all these samples showed low quantum yields (<3%). Notably, the fluorescent quantum yield of P3 in this study was significantly higher than that of reported fluorescent solid polymers (Fig. 5d).

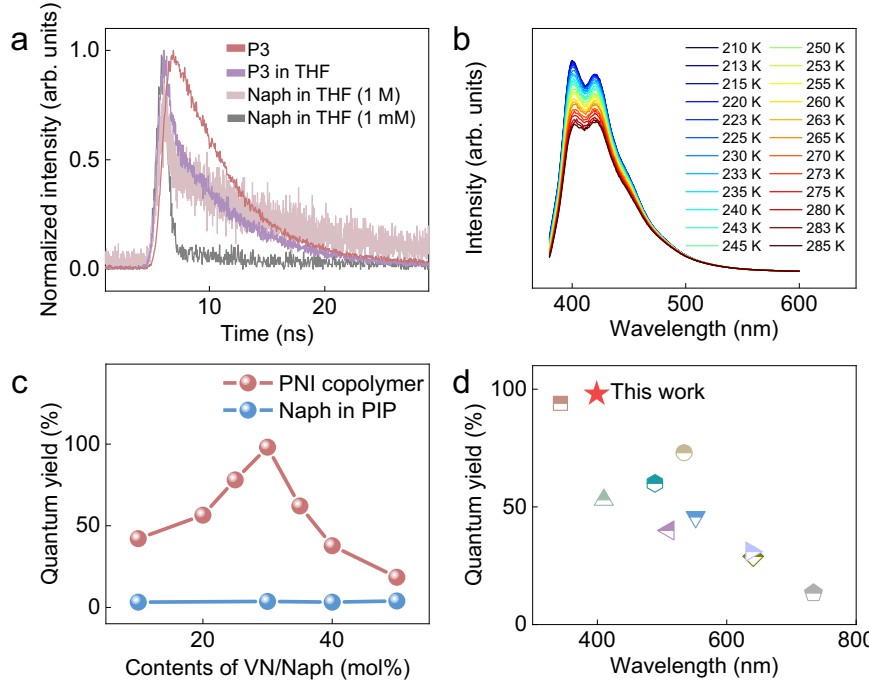

**Fig. 5 | The optical properties of VN-*co*-IP (PNI). a** Fluorescence lifetime analysis. **b** Temperature-dependent fluorescence spectroscopy of P3 (from 210 K to 285 K). **c** PLQY *vs.* VN/Naph content[67,68]. **d** Comparison with literature PLQY values of recent fluorescent polymers[25,34,69–75].

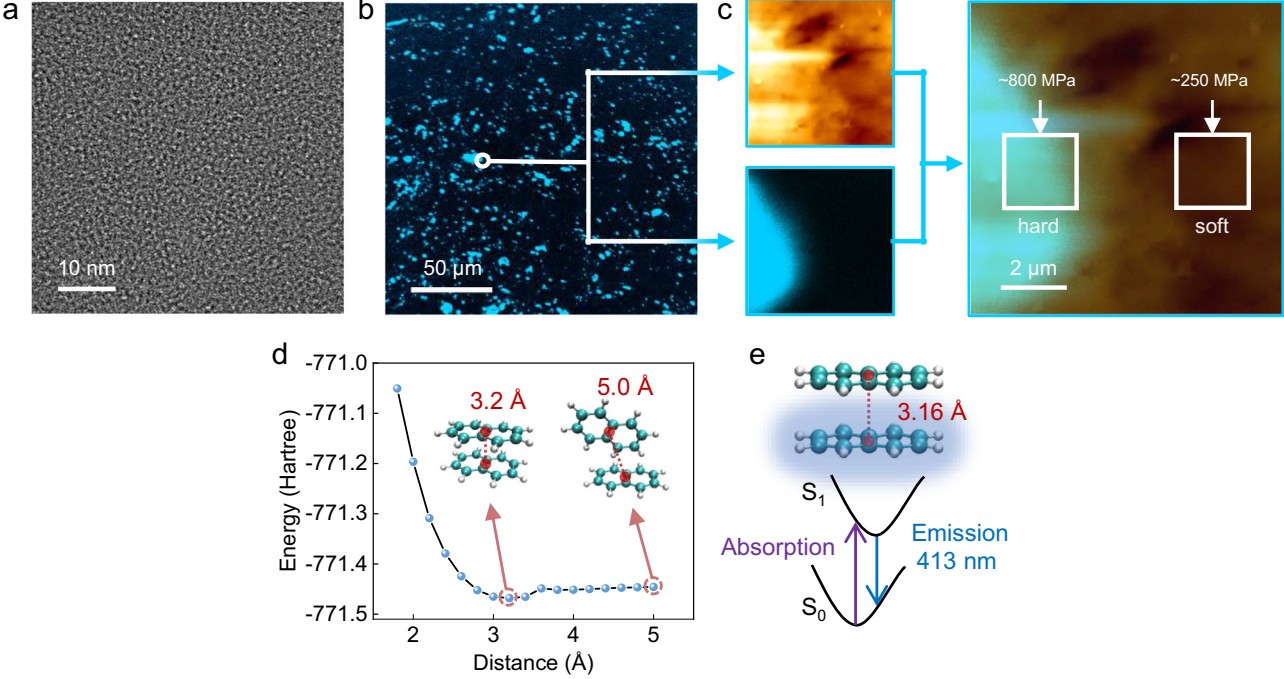

**Fig. 6 | Microstructural characterization and theoretical simulation of VN-*co*-IP (PNI). a** Transmission electron microscopy (TEM) image of P3 (scale bar: 10 nm). **b** Fluorescence confocal microscopy images of P3. **c** Images of in-situ fluorescence confocal microscopy coupled with atomic force microscopy (AFM) of P3, with modulus mapping: rigid region (left, 800 MPa) and soft region (right, 250 MPa). **d** Potential energy surface (PES) scan of the naphthalene excimer. **e** Unrestricted excited-state optimization at the PES minima and proposed fluorescence mechanism.

To further investigate the polymer-constrained excimer structure, the microstructure of P3 was analyzed by using the transmission electron microscope (TEM) and in situ fluorescence confocal microscopy coupled with atomic force microscopy. As shown in Fig. 6a, b, distinct microphase-separated domains were observed in P3. In the selected area of Fig. 6b, the fluorescent regions exhibited relative modulus three times higher than that of non-fluorescent areas, indicating that the fluorescence of the copolymer arises from the hard phase composed of naphthyl-naphthyl interactions, further confirming the formation of polymer constrained excimers in the harder naphthyl-naphthyl domains under UV (Fig. 6c and Supplementary Fig. 19).

To further explore the fluorescent mechanism in more detail, a potential energy surface (PES) scan of the naphthalene excimer has been conducted, as shown in Fig. 6d. It was revealed that as the center-of-mass distance of the excimer in the excited state decreased from 5 Å, the conformational energy initially reduced and then increased, reaching a minimum at 3.2 Å. A shorter center-of-mass distance corresponded to a higher naphthalene concentration. Additionally, using the geometry with the minimum conformational energy in the excited state, we performed an unrestricted excited-state geometry optimization and calculated the fluorescence emission of the naphthalene excimer, as illustrated in Fig. 6e. The optimized center-of-mass distance was 3.16 Å, closely matching the 3.2 Å from the PES scan. The calculated fluorescence emission wavelength of 413 nm aligned well with the experimental value of 396 nm. These computational findings are consistent with experimental observations and provide a clear explanation for the photoluminescence mechanism of PNI copolymers.

## Self-healing fluorescent flexible electrets prepared by PNI copolymer

It was hypothesized that the aggregation of naphthyl groups in a typical aromatic-rich thermoplastic elastomer could impart electret properties, enabling effective charge storage. As shown in Fig. 7a, the corona charging method was used to deposit charges on the surface of the PNI copolymer film with different VN contents by ionizing the air[64]. The surface potential of P6 (201 kDa, 10 mol% VN) decreased rapidly from −1500 V to −12 V in four days, whereas P3 (184 kDa, 27 mol% VN) and P7 (86 kDa, 50 mol% VN) maintained high surface potentials of −766 V and −1039 V, respectively, even after 30 days (Fig. 7b). To clarify the mechanism of charge storage of P3 and P7, the scanning electron microscope (SEM) was used to study their micro morphology. As

shown in Supplementary Fig. 20, there were nearly no pores in the polymer film to store charges, revealing that the stability of the deposited charges only relies on the aggregation of naphthyl groups of PNI copolymer. Compared with P6, copolymers P3 and P7 exhibit enhanced charge storage capacity, which may be attributed to their higher $T_g$s by the higher VN comonomer contents (P6, 10 mol% VN, $T_g = −40\,°C$; P3, 27 mol% VN, $T_g = 11\,°C$; P7, 50 mol% VN, $T_g = 70\,°C$)[65].

Next, a self-healing noncontact single-electrode sensor was fabricated using P3 and a conductive self-healable ionic gel. After the negative polarization of P3 for 30 minutes, it was revealed that P3 exhibited an external electrostatic field due to the deposited charges (Fig. 7c). The surface potential of the metal electrode varied with the distance between the sensor and the metal electrode. The original output voltage was determined to be approximately 6.7 Vpp. After the mechanical damage, the output voltage dramatically decreased to about 3.8 Vpp. When the separated sensor was rejoined and held in contact for a self-healing process for 1 hour, both the PNI copolymer and the conductive ionic gel successfully healed, and the output voltage returned to the original state after healing (Fig. 7d).

A single-electrode sensing system was developed to monitor the motion of a robotic hand. Four sensors were attached to the fingertips of the robotic hand to monitor the bending motion of fingers. As shown in Fig. 7e, a wireless multi-channel measuring circuit was used to measure the output of the sensors and transmit real-time data wirelessly to a cellphone. The movements of robotic fingers can be accurately recorded and displayed through a mobile app by this sensing system. Notably, when exposed to ultraviolet light, the copolymer's fluorescent characteristics enable real-time motion tracking of these robotic fingers - a capability beyond the reach of conventional soft electronics. This feature demonstrates promising applications across

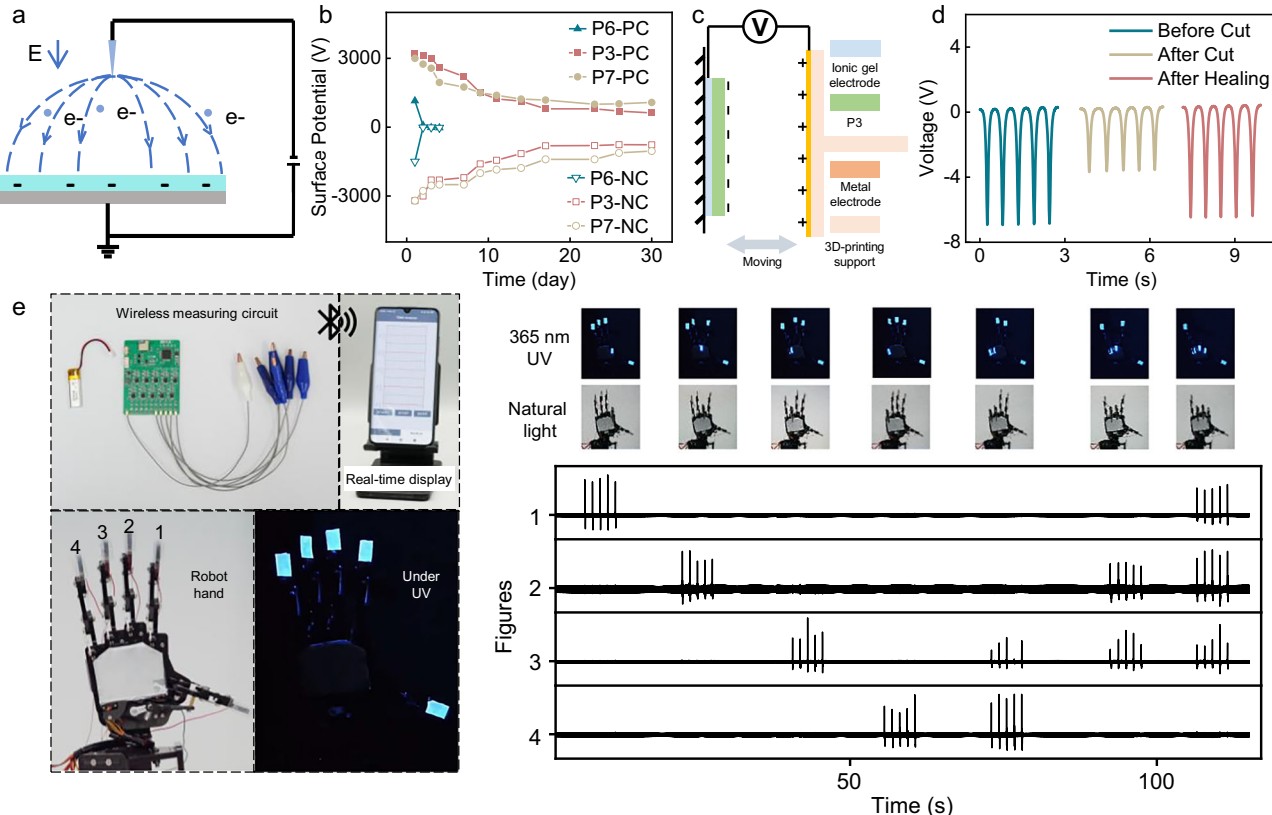

**Fig. 7 | Electret properties and device demonstration of VN-co-IP (PNI). a** Corona polarization setup. **b** Charge retention of the polarized P3, P6, and P7 (PC: positive corona polarization; NC: negative corona polarization). **c** Sensor fabrication. **d** Output voltage: pristine, cut, and healed states. **e** Robotic hand gesture detection under daylight and UV light.

multiple fields such as optically traceable electronic skin, advanced human-machine interfaces (HMI), and deep space exploration systems (Fig. 7e; see also Supplementary Movie 3).

## Discussion

In this work, a family of 1-vinylnaphthalene-co-isoprene copolymers has been synthesized with different microstructures and sequences. By introducing nanoscale naphthalenyl-naphthalenyl microphase separation into the polyisoprene matrix, these copolymers showed the rare combination of mechanical robustness and intrinsic self-healing capabilities at room temperature without the need for external stimuli. The "polymer-constrained excimers" strategy enables the formation of stable naphthyl-naphthyl excimers under UV stimulation, resulting in an ultra-high photoluminescence quantum yield (PLQY > 98%) in the solid state. Furthermore, the material exhibits superior electret performance with more than 30 days of charge retention. Opto-electro-mechanical sensors for the robotic hand made from this material can accurately detect finger movements. The autonomously repairable damage, coupled with exceptional optical and electret properties, positions these copolymers as promising candidates for next-generation wearable technologies. Future work will focus on further optimizing the material's performance and exploring its applications in real-world scenarios.

## Methods

### Materials

Solvents were purified by an MIKROUNA Solv Purer G5 Solvent Purification System and dried over fresh Na chips and molecular sieves in the glovebox (MIKROUNA). 1-Vinylnaphthalene was synthesized according to the literature[66]. Isoprene (IP) and Styrene (St) were purchased from Energy Chemical and purified by distillation from triisobutylaluminium (Al(propyl)$_3$) before use.

1-Naphthaldehyde (97%), Methyltriphenylphosphonium bromide (98%), Potassium tert-butoxide (98%), Isoprene (99%, stabilized with TBC), Styrene (AR, ≥ 99.5%, stabilized with TBC), Methanol (GR, 99.99%; AR, 99.5%), Triisobutylaluminium (1.0 M in $n$-hexane), and sec-Butyllithium (1.3 M $n$-hexane) were obtained from Energy Chemical, Cyclohexane (HPLC, 99.9%) and Tetrahydrofuran (HPLC, 99.9%) were obtained from Yinli. The deuterated solvents Chloroform-d (99.8 atom % D) were obtained from Energy Chemical.

### Synthesis of poly(1-vinylnaphthalene-co-isoprene) copolymer

General polymerization procedure for the synthesis of poly(1-vinylnaphthalene-co-isoprene) copolymers (PNI). Take P3 as an example: In a glovebox, the dried cyclohexane was added into the reaction flask with a magnetic stir bar, followed by the addition of sec-Butyllithium (s-BuLi) solution (diluted tenfold before use, 0.10 mL, 0.013 mmol) with 50 mL cyclohexane. The polymerization was started via the addition of the monomer mixture of 1-vinylnaphthalene (1.60 g, 10.4 mmol) and isoprene (1.77 g, 26.0 mmol), and all the reactions were carried out at room temperature under nitrogen. The living chain ends were terminated by the addition of methanol. To precipitate the polymer, the polymer solution was poured into an 8-fold volume excess of methanol, dried at reduced pressure at 60 °C, and a white solid was obtained (2.76 g, yield 81.9%), and stored in the absence of light at room temperature.

## Data availability

Additional supporting data are available from the corresponding author upon request.

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

## Acknowledgements

This work was supported, in part, by the National Key R&D Program of China (No.2022YFE0103800), the National Natural Science Foundation of China (No.22171091), the Guangdong Basic and Applied Basic Research Foundation (No. 2022A1515011785, 2024 A1515011330), the Introduced Innovative R&D Team of Guangdong (2021ZT09L392), the Pearl River Talent Recruitment Program (2021QN020782), the Science and Technology Program of Guangzhou (2024D03J0003). This work is partially supported by High Performance Computing Platform of South China University of Technology. We are grateful to Professor Zhiguo Xia, Wei Liao, and Zhan Xiong for the measurement of quantum yield. The university logo in Fig. 4A has obtained the usage permission.

## Author contributions

H.W. and S.Z. conceived and designed the experiments. S.Z. performed the synthesis, mechanical, self-healing, and optical experiments, and analyzed the data. D.Z. and J. Z. conceived and designed the electrets application. N.C. and J.Z. conceived computational studies. S.Z., D.Z., N.C., and H.W. prepared the manuscript. S.Z., D.Z., and N.C contributed equally. H.W. directed the project.

## Competing interests

The authors declare no competing interests.
