## [Transparent Peer Review file · Nature Communications]

Polymer-Constrained Excimer Enables Flexible and Self-Healable Optoelectronic elastomer for Mechanical Sensor

Corresponding Author: Professor Haobing Wang

Version 0:

Reviewer comments:

Reviewer #1

(Remarks to the Author)

The paper report the synthesis of a series of co-polymers of vinyl naphthalene (VN) with isoprene. Reactivity studies suggest a random co-polymer, rather than a gradient polymer, as with styrene. The polymers are prepared by anionic polymerisation, and to the best of my knowledge are novel.(various VN polymers have been reported and their excimer effects previously reported, but not these). The polymers are investigated as the VN ratio is changed in terms of their physical and optical properties. Interesting self-healing and fluorescent properties, particularly an outstanding PLQY are reported, which could make the paper suitable for publication in this journal. However, I have some comments and suggestions:

Firstly, the paper is very difficult to follow because nowhere is presented a table explaining what specific samples P1-P5 are. The reader is referred to table S1, which gives some isoprene/vinylnaphthalene ratios, but doesn't specify which is P1-P%, nor NIN-3. PNI is also introduced, with VN content varying, which affects T_g. The various nomenclatures make is very hard to follow the structure property discussion. I suggest the authors include a table in the main paper highlighting what each sample is specifically.

The self-healing properties of the co-polymers are interesting, especially since the 30% VN works well, compared to 10 and 50%. The authors explain this is driven though aggregate formation through pi-pi packing of the naphthalene groups. This appears plausible to explain the poor performance for low VN loadings, but not at high loadings (50%). Here I accept the T_g is higher, but does the polymer self-heal if heated above the T_g? Based on their explanation I would expect this to occur. Exposing the 30%VN polymer to excess soluble naphthalene (methylNP) would also be expected to disrupt the pi-pi aggregation and might therefore inhibit healing/mechanical properties. This could provide further support for their hypothesis (but is not necessary).

My main concern with the paper is the optical properties reported. The paper reports a PLQY (99%) in solid-state films, attributed to excimer emission for one of the VN copolymers. Such an extremely high quantum yield for an excimer is highly unusual in my experience (or indeed for any material), and therefore further support is requested. I also note earlier reports of excimer in (vinylnaphthalene) type polymers or naphthalene have much lower PLQY. From the SI, the PLQY methodology described appears to use a simplified integrating sphere protocol (two-step). I would suggest to use the de Mello method (see also here <https://www.nature.com/articles/s41598-019-51718-4>), which includes a third configuration to correct for scattering, reflection, and background signals. This is particularly important for thin films, as in present case. These are needed for such record claims.

I appreciate that life time measurements help to support the excimer; this could be further supported by temperature-dependent PL (below) T_g, which could help to understand if the excimer formation is dynamic or pre-organised. Similarly since lowering temperature should reduced non-radiative decay, it would help provide further confidence in the very high value quoted.

Reviewer #2

(Remarks to the Author)

This manuscript presents a study on a novel class of polyisoprene-based copolymers incorporating 1-vinylnaphthalene,

resulting in flexible, self-healing elastomers with ultrahigh solid-state photoluminescence quantum yields and electret properties. The findings are supported by both experimental and theoretical analyses. The key innovation is the "polymer-constrained excimer" strategy, which enables solid-state fluorescence within a mechanically robust and self-healable matrix. These properties suggest potential applications in wearable electronics, robotic sensing, and flexible photonic devices. However, several issues need to be addressed prior to publication in Nature Communications.

1. In Figure 3b, only three of the five tensile curves are distinguishable. The authors should either adjust the presentation style or provide individual plots for clarity.
2. In Figure 3e, the images of the original and healed samples don't look like the same sample.
3. The discussions for Figure 3f are confusing. While PNI-50 mol% VN shows limited self-healing in the images, PNI-10 mol% VN exhibits high healing efficiency, which contradicts the current explanation. This discrepancy needs clarification.
4. The proposed explanation for self-healing is overly narrow, attributing self-healing solely to pi-pi interactions and microphase separation. While these factors likely contribute, the role of chain mobility (as indicated by T_g in Figure 3a) and covalent entanglement should also be acknowledged. Comparisons with materials of differing rigidity should be avoided without appropriate context.
5. Use consistent font styling in Figure 5 d-e with others.
6. For charge retention results in Figure 6b, the reason for varying initial surface potential is unclear. If charge retention is linked to naphthyl group aggregation in copolymers with higher VN content, why is there minimal difference between P3 and P7? The microstructural features of P6 and P7 should be discussed?
7. While the sensor design is intriguing, the manuscript does not clearly justify the necessity or added value of combining optoelectronic and fluorescent properties. The authors should elaborate on the specific applications and advantages over conventional soft electronics and ionotronics.
8. The sensor operation (shown in the video) exhibits significant noise, which varies during use. This issue should be addressed.
9. The signal values in the video do not align with those in Figure 6e. The authors should clarify whether the data has been normalized or otherwise processed.
10. The finger-sensing demonstration only shows a single type of movement (back-and-forth). The authors should investigate whether the material responds to different types of motion and varying degrees of the same motion.

Reviewer #3

(Remarks to the Author)

In this study, Zheng and coauthors report self-healable materials showing photoluminescent properties. By employing a "polymer-constrained excimer" strategy, these copolymers exhibit remarkable photoluminescent properties, achieving an ultra-high photoluminescence quantum yield (PLQY > 99%). The simultaneous achievements of both self-healing properties and photoluminescence is impressive and the mechanism is well discussed. The reviewer recommends publication after addressing the following comments appropriately.

1. In the introduction, the necessity for self-healing optical and electrical materials is broadly discussed. However, the important results of this research are the luminescence performance and electret properties, thereby the reviewer feels that specific current status and challenges of past research on these are not properly mentioned. These discussion should be appropriately added.
2. The self-healing and luminescence performance of the P3 materials have been evaluated independently, but there is a lack of data to show the relationship between the two properties. The reviewer recommends additional experiments to evaluate the recovery of luminescence performance after self-healing. Specifically, the fluorescence intensity and/or photoluminescence quantum yield (PLQY) of the repaired samples should be measured and compared to the undamaged samples.
3. The caption for Figure 1 is unclear and difficult to understand. A more detailed caption is needed for each panel. Each panel (a-c) must be specifically referred in the main text.
4. On page 6, "being cut and repaired for 30 min": It must be described more precisely because it is quite unclear. In the experimental section, there is no description of the duration of "30 min". The reviewer guesses the sample was stored for "30 min." after "the cut faces were brought together and gently pressed for less than 15 seconds at 25 °C". These must be specified.
5. On page 6, "When the sample was cut and repaired at 36 °C": Does this mean that the "the cut faces were brought together and gently pressed for less than 15 seconds at 36°C, instead of 25 °C? Was the temperature during storage (6 h) also increased to 36°C?

Version 1:

Reviewer comments:

Reviewer #1

(Remarks to the Author)

I'm happy with the changes, and the authors have addressed all my comments clearly. I think the paper can now be accepted.

Reviewer #2

(Remarks to the Author)

My concerns about the manuscript have been revised in detailed, it's now recommended for publication.

Reviewer #3

(Remarks to the Author)

I'm pleased to see that the authors addressed all the comments appropriately. This paper is now ready for publication.

Response to Reviewers

Reviewer: 1

Recommendation: *Interesting self-healing and fluorescent properties, particularly an outstanding PLQY are reported, which could make the paper suitable for publication in this journal.*

Comments:

The paper report the synthesis of a series of co-polymers of vinyl naphthalene (VN) with isoprene. Reactivity studies suggest a random co-polymer, rather than a gradient polymer, as with styrene. The polymers are prepared by anionic polymerisation, and to the best of my knowledge are novel.(various VN polymers have been reported and their excimer effects previously reported, but not these). The polymers are investigated as the VN ratio is changed in terms of their physical and optical properties. Interesting self-healing and fluorescent properties, particularly an outstanding PLQY are reported, which could make the paper suitable for publication in this journal. However, I have some comments and suggestions.

Response:

We sincerely appreciate the reviewer's evaluation of this work.

Comment:

1) the paper is very difficult to follow because nowhere is presented a table explaining what specific samples P1-P5 are. The reader is referred to table S1, which gives some isoprene/vinylnaphthalene ratios, but doesn't specify which is P1-P%, nor NIN-3. PNI is also introduced, with VN content varying, which affects Tg. The various nomenclatures make is very hard to follow the structure property discussion. I suggest the authors include a table in the main paper highlighting what each sample is specifically.

Response:

Thank you very much for your comment. We have added a new table for the preparation of **P1 to P5** in the revision manuscript. See Table 1.

Comment

2) The self-healing properties of the co-polymers are interesting, especially since the 30% VN works well, compared to 10 and 50%. The authors explain this is driven though aggregate formation through pi-pi packing of the naphthalene groups. This appears plausible to explain the poor performance for low VN loadings, but not at high loadings (50%). Here I accept the Tg is higher, but does the polymer self-heal if heated above the Tg? Based on their explanation I would expect this to occur.

Response:

When the VN content is below 40%, the glass transition temperature (T_g) of the material remains below room temperature, enabling effective self-healing under ambient conditions. However, at a VN content of 50%, the material enters a glassy state at room temperature and loses its self-healing capability. In this case, when damage occurs, raising the temperature to 60°C restores excellent self-healing performance.

Comment

3) Exposing the 30% VN polymer to excess soluble naphthalene (methylNP) would also be expected to disrupt the pi-pi aggregation and might therefore inhibit healing/mechanical properties. This could provide further support for their hypothesis (but is not necessary).

Response :

Following the reviewer's suggestion, we evaluated the addition of methylNP to the 30% VN polymer formulation. This modification resulted in undesirable material properties:

The doped material hardened into a plastic-like state, and a significant reduction in mechanical performance was observed. As the reviewer expected, this material completely lost room-temperature self-healing capability.

Comment:

4) My main concern with the paper is the optical properties reported. The paper reports a PLQY (99%) in solid-state films, attributed to excimer emission for one of the VN copolymers. Such an extremely high quantum yield for an excimer is highly unusual in my experience (or indeed for any material), and therefore further support is requested. I also note earlier reports of excimer in (vinyl)naphthalene type polymers or naphthalene have much lower PLQY. From the SI, the PLQY methodology described appears to use a simplified integrating sphere protocol (two-step). I would suggest to use the de Mello method (see also here <https://www.nature.com/articles/s41598-019-51718-4>), which includes a third configuration to correct for scattering, reflection, and background signals. This is particularly important for thin films, as in present case. These are needed for such record claims.

Response:

According to the reviewer's suggestion, we have recalculated all fluorescence quantum yields using the de Mello method. The revised maximum fluorescence quantum yield has been adjusted from >99% to 98.1%. The corresponding calculation method reference has been properly cited in the revised manuscript (see ref. 64 and 65). We sincerely appreciate the reviewer's valuable suggestion, which has significantly improved the accuracy of our data.

Comment:

5) I appreciate that life time measurements help to support the excimer; this could be further supported by temperature-dependent PL (below) T_g , which could help to understand if the excimer formation is dynamic or pre-organized. Similarly since lowering temperature should reduced non-radiative decay, it would help provide further confidence in the very high value quoted.

Response:

According to the reviewer's suggestion, we have conducted temperature-dependent photoluminescence (PL) studies on **P3**, with detailed analysis presented in Figure 5b. The PL measurements reveal that the excimer formation in **P3** occurs through a pre-organized mechanism rather than a dynamic process. This conclusion is supported by the absence of additional red shift in the emission spectrum below the glass transition temperature ($T_g = 11$ °C).

The description has also been added in Page 9, right column: The temperature-dependent fluorescence spectroscopy of **P3** also showed a significant fluorescence intensity increase as the temperature decreased (Figure 5b) due to the further reduction of non-radiative decay; these two phenomena further confirmed the existence of stable excimers in the solid state.

Reviewer: 2

Recommendation: Several issues need to be addressed prior to publication in Nature Communications.

Comments:

This manuscript presents a study on a novel class of polyisoprene-based copolymers incorporating 1-vinylnaphthalene, resulting in flexible, self-healing elastomers with ultrahigh solid-state photoluminescence quantum yields and electret properties. The findings are supported by both experimental and theoretical analyses. The key innovation is the “polymer-constrained excimer” strategy, which enables solid-state fluorescence within a mechanically robust and self-healable matrix. These properties suggest potential applications in wearable electronics, robotic sensing, and flexible photonic devices. However, several issues need to be addressed prior to publication in Nature Communications.

Response:

We sincerely appreciate the reviewer’s evaluation of this work.

Comment:

1) In Figure 3b, only three of the five tensile curves are distinguishable. The authors should either adjust the presentation style or provide individual plots for clarity.

Response:

Thank you very much for your comment. We have revised Figure 3b to make it more distinguishable.

Comment

2) In Figure 3e, the images of the original and healed samples don’t look like the same sample.

Response:

These two images are indeed of the same sample. The observed differences resulted from variations in shooting conditions, including camera lenses and light intensity, as the photos were taken at different times. In the revised manuscript, we have re-shot the demonstration of the sample's self-healing properties (see Figure 3e for updated results).

Comment

3) The discussions for Figure 3f are confusing. While PNI-50 mol% VN shows limited self-healing in the images, PNI-10 mol% VN exhibits high healing efficiency, which contradicts the current explanation. This discrepancy needs clarification.

Response :

Thank you very much for your comment.

The corresponding description has been updated. See Page 6, right column: **standing in sharp contrast with the VN ratios of 50 mol%, which presented no self-healability under the same conditions. As for PNI with 10 mol%, the sample showed limited self-healability and was not able to completely heal the wound in 30 min.**

Comment:

4) The proposed explanation for self-healing is overly narrow, attributing self-healing

solely to pi-pi interactions and microphase separation. While these factors likely contribute, the role of chain mobility (as indicated by T_g in Figure 3a) and covalent entanglement should also be acknowledged. Comparisons with materials of differing rigidity should be avoided without appropriate context.

Response:

Thank you very much for your suggestion. The explanation for self-healing has been proposed in the revised manuscript. See Page 7: The exceptionally rapid self-healing may result from the high mobility of the *cis*-1,4-polyisoprene segments with low T_g , which could induce the rapid re-aggregation of the naphthyl groups through π - π interaction to repair the mechanical damage.

Comment:

5) Use consistent font styling in Figure 5 d-e with others.

Response:

Thank you very much for your comment.

In response to the reviewer's comments, we have revised the style of Figure 5 d-e.

Comment:

6) For charge retention results in Figure 6b, the reason for varying initial surface potential is unclear. If charge retention is linked to naphthyl group aggregation in copolymers with higher VN content, why is there minimal difference between P3 and P7? The microstructural features of P6 and P7 should be discussed?

Response:

To systematically investigate the influence of vinyl naphthalene (VN) comonomer content on charge storage capacity, we evaluated copolymer **P8** (containing 20 mol% VN, $T_g = -20$ °C), which exhibited poor charge retention properties, losing nearly all stored charges within 15 days. These results demonstrate a strong correlation between charge storage performance and both VN content and glass transition temperature (T_g) in these copolymers. See ref. 74, *J. Appl. Polym. Sci.* **27**, 1107-1118 (1982).

TEM analysis revealed no significant microstructural differences between **P3** and **P7**, suggesting that the naphthalene groups in **P3** are already sufficient for effective electret charge stabilization. This explains why both **P3** and **P7** demonstrate comparable charge storage capabilities despite their compositional differences.

Modifications made to the revised manuscript include: P12, right column, Compared with **P6**, copolymers **P3** and **P7** exhibit enhanced charge storage capacity, which may be attributed to their higher T_g s by the higher VN comonomer contents (**P6**, 10 mol% VN, $T_g = -40$ °C; **P3**, 27 mol% VN, $T_g = 11$ °C; **P7**, 50 mol% VN, $T_g = 70$ °C).⁷²

Comment:

7) While the sensor design is intriguing, the manuscript does not clearly justify the necessity or added value of combining optoelectronic and fluorescent properties. The authors should elaborate on the specific applications and advantages over conventional soft electronics and ionotronics.

Response:

Thank you for your suggestion.

The potential application and advantages of this material have been discussed in the revised manuscript. See Page 13, right column: Notably, when exposed to ultraviolet light, the copolymer's fluorescent characteristics enable real-time motion tracking of these robotic fingers - a capability

beyond the reach of conventional soft electronics. This unique feature demonstrates promising applications across multiple fields such as optically traceable electronic skin, advanced human-machine interfaces (HMI), and deep space exploration systems (**Figure 7e**; see also **Supplementary Video 3**).

Comment:

8) *The sensor operation (shown in the video) exhibits significant noise, which varies during use. This issue should be addressed.*

Response:

Thank you very much for your comment.

Due to the laboratory-built nature of our circuit boards, complete electromagnetic interference (EMI) shielding could not be achieved, resulting in observable signal noise. Despite multiple optimization attempts, we were unable to effectively reduce these noise signals to a negligible level. We have confirmed that this noise does not affect the evaluation of the material's electret response.

Comment:

9) *The signal values in the video do not align with those in Figure 6e. The authors should clarify whether the data has been normalized or otherwise processed.*

Response:

Thank you very much for your comment.

In Figure 7e, we initially presented data with both standardized inter-gesture intervals and low-pass filtering. For the revised manuscript, we maintain the standardized timing protocol while providing the unfiltered dataset to enable direct evaluation of the raw signal characteristics.

Comment:

10) *The finger-sensing demonstration only shows a single type of movement (back-and-forth). The authors should investigate whether the material responds to different types of motion and varying degrees of the same motion.*

Response:

Thank you very much for your comment.

This study primarily focuses on the synthesis, characterization, and preliminary applications of a novel polymeric material system exhibiting self-healing capability, exceptional mechanical properties, ultrahigh fluorescence quantum yield, and electret characteristics. Current ongoing research is investigating more sophisticated responsive behaviors of these materials, including their performance under various motion modes for advanced applications.

Reviewer: 3

Recommendation: The reviewer recommends publication after addressing the following comments appropriately.

Comments:

In this study, Zheng and coauthors report self-healable materials showing photoluminescent properties. By employing a "polymer-constrained excimer" strategy, these copolymers exhibit remarkable photoluminescent properties, achieving an ultra-high photoluminescence quantum yield (PLQY > 99%). The simultaneous achievements of both self-healing properties and photoluminescent is impressive and the mechanism is well discussed. The reviewer recommends

publication after addressing the following comments appropriately.

Response:

We sincerely appreciate the reviewer's evaluation of this work.

Comments:

1) *In the introduction, the necessity for self-healing optical and electrical materials is broadly discussed. However, the important results of this research are the luminescence performance and electret properties, thereby the reviewer feels that specific current status and challenges of past research on these are not properly mentioned. These discussions should be appropriately added.*

Response:

Thank you very much for your suggestion. We added the current status and challenges of electret materials in the revised manuscript. See Page 2, right column: **Aromatic-containing polymers also constitute an important category of polymer electrets...**

Comments:

2) *The self-healing and luminescence performance of the P3 materials have been evaluated independently, but there is a lack of data to show the relationship between the two properties. The reviewer recommends additional experiments to evaluate the recovery of luminescence performance after self-healing. Specifically, the fluorescence intensity and/or photoluminescence quantum yield (PLQY) of the repaired samples should be measured and compared to the undamaged samples.*

Response:

We appreciate the reviewer's comment. As suggested, we have added the experiments to evaluate the recovery of luminescence performance of **P3** after self-healing, and the corresponding revisions have been incorporated in the manuscript. See Page 8, right column: **We investigated the fluorescence behavior during the self-healing process of P3 (Supplementary Fig. 16). Immediately after cutting and rejoining the P3 specimen, we observed a significant fluorescence enhancement. This phenomenon likely results from increased light absorption at the freshly exposed wound surfaces compared to intact regions, where excitation light was attenuated by the penetration process and thus generated less fluorescence. As self-healing progressed, the fluorescence intensity gradually decreased, eventually returning to near-baseline levels upon complete restoration of the material.**

Comments:

3) *The caption for Figure 1 is unclear and difficult to understand. A more detailed caption is needed for each panel. Each panel (a-c) must be specifically referred in the main text.*

Response:

Thank you very much for your suggestion. A very detailed caption has been updated in the revised Figure 1.

Comments:

4) *On page 6, "being cut and repaired for 30 min": It must be described more precisely because it is quite unclear. In the experimental section, there is no description of the duration of "30 min". The reviewer guesses the sample was stored for "30 min." after "the cut faces were brought together and gently pressed for less than 15 seconds at 25 °C". These must be specified.*

Response:

According to the reviewer's comments, the corresponding revisions have been incorporated into the manuscript. See Page 6, left column: For instance, when a dog-bone-shaped film sample of P3 was completely cut by a razor blade, the material could be stretched to 550% after the cut area was contacted and gently pressed for less than 15 seconds at 25 °C in air, and then allowed to self-repair for 30 min.

Comment:

5) On page 6, “When the sample was cut and repaired at 36 °C”: Does this mean that the “the cut faces were brought together and gently pressed for less than 15 seconds at 36°C, instead of 25 °C? Was the temperature during storage (6 h) also increased to 36°C?

Response:

We appreciate the reviewer's comment.

We make the following change in Page 6, right column: When the sample was cut, contacted, and placed under 36 °C (human body temperature) in air, a complete recovery was observed within 6 h, indicating its high potential for self-healing wearable devices or electronic skin.